# Mobility to Other Locations: A Study on the Spread of the Cult of Lord Yan from Jiangxi to Hubei in the Ming–Qing Era

**Shuaiqi Zhang [1],\* and Hongyu Sun [2],\***

1   School of History, Wuhan University, Wuhan 430072, China
2   School of Sociology, Wuhan University, Wuhan 430072, China
\*   Correspondence: zsq0912@whu.edu.cn (S.Z.); sunhongyu@whu.edu.cn (H.S.)

**Abstract:** In the Yuan Dynasty, Lord Yan 晏公 was worshipped by the people of Jiangxi 江西 as a water god, but there was no consensus on the identity of the god and the process of his deification. During the transitional period between the Yuan and Ming dynasties, the cult of Yan Gong was increasingly popular among different social groups in the Qingjiang 清江 region. Later, thanks to a combination of officials, merchants, and immigrants, its spatial scope was extended to Hubei 湖北 Province. During the Hongwu 洪武 (r. 1368–1398) period, the cult of Lord Yan in Hubei was so prevalent that multiple groups of people were enthusiastically involved in the construction of Lord Yan temples; thus, many temples shot up along lakes and the main tributaries of the Yangtze River, constituting a geographical distribution pattern with a concentration in the central and eastern parts and a scarcity in the west. The reason for this was the multidimensional interaction of migration activities, the cross-regional economic activities of merchants, and the promotion of folk beliefs by local officials since the Ming–Qing era, which encompasses the historical evolutionary features of actors competing for the cult of gods and control of regional social power.

**Keywords:** Ming–Qing era; cult of Lord Yan; community worship; spatial expansion

## 1. Introduction

The mobility of local deities beyond their borders is a basic feature of the historical evolution of folk beliefs. "Mobility" is both a subjective choice of the people in folk beliefs and a realistic manifestation of the intended utility of the cult (Z. Zhang 2015, pp. 188–203). It has greatly enhanced the spatial influence of deities, which is indispensable for the presentation of folk culture, the communication of regional ties, and insight into social changes. Many social classes or collective organizations have different or contradictory interpretations of the deities, but they have accumulated and cross-fertilized over a long period of time to create an image of divine authority (Duara 2010, p. 112; Johnson et al. 1985, pp. 93–114).

The cult of Lord Yan is the cultural bearer of the local community's thought in Jiangxi, as well as the "Ritual Signs" 禮儀標識, which integrate the sense of community identity and reveal the daily life of people (Faure 2016, pp. 24–27, 158). The concept of the cult of gods embedded in them is always under dynamic construction due to temporal and spatial changes or class consciousness.

The local community has an independent social order, and the ideological order is an integral part of it (Sen 2017, p. 40). Additionally, the cult of God is an element in the local social ideology order, which is the cultural connection between ideology and social groups. Therefore, the emergence, evolution, and spread of the cult of Lord Yan cannot be separated from the Jiangxi community, yet the influence of different social groups and pre-existing ideological orders are equally significant. The cult of Lord Yan originated in Qingjiang Town 清江鎮, Linjiang Prefecture 臨江府, Jiangxi Province. Later, with the

passage of time and the interweaving of multiple factors, it gradually divorced from traditional principles, crossed the original geographical and spatial boundaries of "Living in a Remote Area" 僻處一隅, and was widely worshiped as a water deity by civil society.

As the transport hub of central China, Hubei enjoys geographical privilege, with the Yangtze River running through, five streams interlocking, the Han River as the bond, and the Mountain of Heng 衡山, Lake Dongting 洞庭湖, and Marsh of Yunmeng 雲夢澤 as guards (Lv 1934, p. 1). Neighboring Jiangxi Province, with rivers and lakes intertwined, naturally became an important field for the spread of the cult of Lord Yan.

Based on previous research,[1] this paper takes the local chronicles and circulating documents of Jiangxi and Hubei provinces as the basic historical materials, trying to explain the changes in the divine character of the Jiangxi Lord Yan, the spatial layout and expansion of Lord Yan's temples in Hubei Province, and the state of the cult of Lord Yan by social groups during the Ming and Qing dynasties. Furthermore, it reveals the characteristics of the time when the state's will, regional migration flows, spatial commercial interactions, and social pluralistic groups were intermingled.

## 2. The Legend of Apotheosis and the Evolution of the Cult of Lord Yan

The cult of Lord Yan originated in Qingjiang County, Linjiang Circuit 臨江路, Jiangxi Province, during the Yuan Dynasty, and the local communities deified the folklore and shaped the deity of refuge.

*Chu Yin Ji* 樗隱集 (*Collected Works of Chuyin*),[2] the first private collection of documents documenting the ancestral temple of Lord Yan, was written by Hu Xingjian 胡行簡 (fl. 1354) of the Yuan dynasty. The collection contains the contents of the Inscription of Lord Yan Temple in Qingjiang Town 清江鎮晏公祠廟碑. There is a Lord Yan Temple in Qingjiang Town, which has passed through many dynasties, but the overall appearance has not changed. People visiting to offer sacrifices found the temple was narrow and shabby and wanted to demolish and renovate it, but such effort was fruitless. In 1385, countryman Peng Shikuan 彭士寬 (fl. 1385–1390) said, "Lord Yan is the protector of the local people, but the modest appearance of the temple is not enough to prove the people's devotion to Lord Yan, so isn't this a lack of ceremony?" When the people who had been blessed by Lord Yan heard his words, they donated either money or food to help renovate the temple. Therefore, the land on the side of Mount Baojin 寶金山 featured the rebuilding of new temples and corridors, which were several times larger in size and quantity than the original ones (Hu 1982, p. 154).

The contents of the inscription indicate that Lord Yan Temple existed in the Linjiang area of Jiangxi Province during the Yuan dynasty, but records of the large-scale construction of Lord Yan Temple during the Hongwu period of the Ming Dynasty were found in an inscription of the Yuan Dynasty and raise doubt about the authenticity of the statement that the cult of Lord Yan began in the Yuan dynasty. However, this fact was mostly recorded in prefecture and county annals of multiple regions; for example, "Yan Wuzai 晏戌仔 (the full name of Lord Yan), born in Qingjiang Town at the end of the Song Dynasty, was Head of the Hall of the Wenjin Bureau 文錦局" (Pan 1975, p. 24)—considering this, it is difficult to deny the statement. Yan Wuzai was mainly active in the late Song and early Yuan dynasties and later became a local water deity worshiped by the people of Qingjiang County. Therefore, the Cult of Lord Yan was generated no later than the end of the Yuan dynasty.[3]

Lord Yan's life story and deification process have had various versions since the Yuan–Ming era, and the reason for this is that the legend of the deification of folk society has diversified characteristics (Puett 2002, pp. 245–95; Ter Haar 2000, pp. 451–55).

The book *Huitu Sanjiao Yuanliu Soushen Daquan Wai Erzhong* 繪圖三教源流搜神大全外二種 (*Sources and Images of the Three Religious Deities and Two Other Books*), written in the Ming Dynasty, records that Lord Yan died and became a god. At the beginning of the Yuan Dynasty, Lord Yan was elected as an official due to his talent, holding office as Head of the Hall of the Wenjin Bureau. He died due to illness as soon as he boarded the boat returning to his hometown, and the subordinates put his body in the coffin as a ritual. Before the

boat arrived at his hometown, the people saw Lord Yan on his horse, galloping across the fields and wearing the same clothes as usual. A month had passed by the time Lord Yan's body finally arrived. People were shocked to hear the news that Lord Yan had died on the day they saw him. When they opened the coffin to look, there was nothing inside. Elderly people knew what that meant and set up a temple to worship him (Anonymous 2012, pp. 400–1).

However, Luo Maodeng 羅懋登 (fl. 1396) in the Ming Dynasty recorded in his *Taijian Sanbao Xiyang Ji Tongsu Yanyi* 太監三寶西洋記通俗演義 (*Eunuch Sambo's Overseas Travels*) that Lord Yan was apotheosized due to his good deeds to help the people. The Yuan government was tyrannical, and people were undergoing heavy taxation. The officials of the Wenjin Bureau were responsible for managing the supply of brocade to the court. A worker, Pu Er 濮二 (fl. 1325), who was unable to afford to weave, had to sell his son and two daughters to compensate higher officials. Lord Yan took pity on him for his experience and gave his salary to support him, but the money was insufficient, so Lord Yan sold his wife's earrings and hairpin for the rest of the money. Because of him, the Pu family was reunited and prayed for divine protection. God valued Lord Yan's integrity and honesty and, therefore, appointed him as the god of water (Luo 1985, p. 1259).

Both of the folklores use "corpse dissolution visions" as the externalized form of expressing the divinity of Lord Yan, but the latter is more likely to satisfy the people's spiritual expectation of "good deeds-reward", which is in line with the moral compliance of ancient society.

There is a common metaphor in traditional Chinese culture that "people who fulfill their filial duties become gods", which is in line with the expression of the folk system of cultivating the immortal gods—"those who want to seek immortality should be based on loyalty, filial piety, harmony and benevolence". The contents of the *Temple Tablet of Lord Yan Ancestral Temple in Qingjiang Town* recorded that "Lord Yan was born with a unique quality, taking good care of his parents and people spoke highly of his filial piety, he was born a filial son, died a bright god." (Hu 1982, p. 154.)

The official enthronement is an important part of the national legal recognition and folk society cult of local deities, but this must be based on the premise of having "righteous deeds in life" and being able to "apparitions after death" (Hamashima 2008, p. 88); *Qi Xiu Lei Gao* 七修類稿 (*A Collection of Seven Types of Literary Novels*) has a similar formulation. In the early days of the Ming Dynasty, the river bank often collapsed because "Zhu po long" 豬婆龍 was digging up the river banks. His name is pronounced in the same way as the surname of the Ming emperor, and thus, it was blamed on "Yuan" 黿. Meanwhile, because his name has the same pronunciation as the Yuan dynasty, rulers ordered that officials catch all "Yuan". However, the river bank collapsed as before. An old fisherman passed and said, "you may use a roast pig as bait to catch it". Then, the people did as he said, but they did not have enough power to catch it. The next day, the old fisherman again said, "the Yuan uses its four feet to climb the rocks, so you should put an 'Urn' connected with fishing cord under its bottom and put it down until the 'Urn' covers its head". Then, it must use its two front feet to resist; thus, you could take control with a combined effort to get its feet up, and the advice was true. The crowd said, "it is a Yuan". The old fisherman said, "the Yuan is so huge that it can swallow people, so we nickname it the 'Zhu Po long'. You can tell the emperor and the river bank can be saved". The crowd asked for his name, and he answered, "Yan is my surname", then abruptly disappeared. After the shore was built, Ming Taizu 明太祖 (r. 1328–1398) realized and said, "it was Lord Yan who saved me from the overturned boat". So, he conferred the titles of Governor Grand Marshal of the Divine Sky Yufu to Lord Yan and ordered officials to worship him (Lang 2009, p. 128).

The statement that Lord Yan assisted in the control of the river disaster and was recalled by Ming Taizu for rescuing the overturned boats and was conferred the title of God was adopted by the Ming Dynasty's Wang Qi 王圻 (1530–1615) in *Baishi Huibian* 稗史彙編 (*A Compilation of Folklore or Old Street Stories*) (Q. Wang 1993, p. 2045), highlighting the aim of local communities to join the ruling class to realize the legitimacy of regional deities.

By the time Zhao Yi 趙翼 (1727–1814) in the Qing dynasty studied the origin of Lord Yan Temple at Baiyun Ferry in Changzhou City 常州城, he had already read the *Qi xiu lei gao* to understand the story; thus, the content of Lord Yan's becoming God in his book *Gai Yu Cong Kao* 陔余叢考 (*Reading Notes on the Free Time of Supporting Parents*) mainly follows this story. Furthermore, he gave a detailed description of Lord Yan's rescuing Ming Taizu (Zhao 2007, p. 728).

However, what he later wrote in *Yan Bao Za Ji* 簷曝雜記 (*A Collection of Fragmented Records from Different Living Areas*) is slightly different from that in the previous book, "People in the past thought that the coir rope monster in the river stricken by Xu Jingyang's 許旌陽 (239–374) magical seal became God" (Zhao 1982, p. 116). Lord Yan was conferred the title of God due to his involvement in defeating the Yuan, Zhu Yuanzhang recalled of Lord Yan saving his life, but it does not mention anything about the coir rope monster. However, the saying that "the coir rope was transformed into a god" was not without origin, but probably originated from *Qi Yin* 齊音 (*Poems in Praise of Jinan Mountains, Lakes and Springs*), written by Wang Xiang Chun of the Ming Dynasty. "The folklore goes that there were two coir ropes named Zong Number One and Zong Number Two residing in the river as monsters. They could not become gods, so they were not able to receive any sacrifice. Xu Jingyang crossed the river, eating persimmons and throwing the rest into the river. The two Zongs with persimmons as eyes, approached Xu, stopped his boat, and showed up in front of him. Xu had no way to defend himself in such a sudden but took the magical seal and hit them in the forehead. The two zongs got the seal and became gods, one called Lord Yan and the other called Xiao Gong 蕭公, and receive sacrifice everywhere." (X. Wang 1993, p. 64.)

*Tianhou Xiansheng Lu* 天后顯聖錄 (*The Book of the Tianhou Concubine*), written by Ming people, also recorded that Lord Yan was "transformed into god", but this was achieved by Tian Fei 天妃 rather than Master Xu Jingyang. There was a river monster called "Lord Yan". As he approached, Tian Fei ordered him to cast down a rope to draw his attention and tied him up without his awareness. Lord Yan, floating on the river, began to experience fear and confessed his guilt. Tian Fei then said, "The East China Sea is full of difficulties and dangers. You are now one of the water gods in my cabinet, and you should protect the people in danger". (Anonymous 2014, pp. 359–60.) Xu Jingyang and Tian Fei lived in the Jin and Song dynasties, but the theory of the "transformation of god" was mostly created by Ming people. The statement "Lord Yan was transformed into god" is absurd, and its authenticity cannot be confirmed.

The legends of Lord Yan becoming a god, as seen in Yuan and Ming literature, vary, with "becoming a god after death", "doing good deeds to become a god", "becoming a god because of filial piety", "becoming a god for helping the emperor", and "becoming a god through enlightenment" being common in folk society, and even the identity of Lord Yan is different,[4] which highlights the arbitrariness and complexity of the process of creating gods in ancient Chinese society.

With Lord Yan's deification process being full of mystery, the deeds listed are diverse and unreal, and the reason why the people of Qingjiang society still believe in the legend and worship Lord Yan may have something to do with the natural environment of Qingjiang County. Qingjiang County is located at the intersection of the north–south waterway, which is the only way to Guangdong and Hunan provinces (Zhang 1989, pp. 188–89). The entire area controls the upper reaches of the provincial capital and is a hinterland with dangerous waterways. "Yuan River 袁水 and Gan River 贛江 merge and converge to flow eastwards, but cannot be released, thus the river continues to wash over the western bank" (De 1970, p. 44), and regional floods are frequent, which objectively lays the environmental foundation for the generation and spread of the cult of Lord Yan. Because the cult of Lord Yan catered to the people's psychology of praying to the gods to bless the safety of waterways and quell local water hazards, the deeds of "protecting people's oars from danger" were most common in lakes and rivers.

Combined with existing historical documents, it can be considered that the Qingjiang Town of Linjiang Prefecture after the Yuan Dynasty formed the cult of Lord Yan, but the divine deeds and the title received are unknown, which may be related to the fact that Lord Yan did not yet form into a god. In the late Yuan and early Ming dynasties, Lord Yan was increasingly active in the rivers, lakes, and seas, which attracted the attention of local officials, gentries, and scholars. Additionally, the rulers repeatedly offered words of reward to provide a legal basis for Lord Yan to become the orthodox deity of the state. However, this shows that the national cult of Lord Yan as a deity was formed in the early Ming Dynasty (Yang 2022).

## 3. Lord Yan's Worship and the Construction of Temples in Hubei Province

According to the above discussion, the legend of Lord Yan can be traced back to the end of the Song Dynasty, but his true identity is still unknown. A more credible identity of Lord Yan is Yan Wuzai, who lived in the Song–Yuan era. Meanwhile, there is no certainty as to when Lord Yan became a god, but what is certain is that he was worshipped by social groups as a water god. Moreover, we can be sure that the cult of Lord Yan originated in Qingjiang Town, Linjiang Circuit, Jiangxi Province, during the Yuan Dynasty, and within this period, it did not spread to other areas because its influence was very limited. However, in the early Ming Dynasty, the cult of Lord Yan began to spread to other regions, driven by the combined efforts of different social groups, such as emperors, officials, gentries, merchants, and civilians (Wang 2020, pp. 105–14), and Lord Yan eventually became a nationally renowned water deity.

The process of spreading any folk beliefs to other regions is influenced to varying degrees by political, economic, cultural, and social factors (Pi 2008, pp. 208–23). At the same time, folk beliefs need the support of the rulers (emperors and officials) and inclusion in the list of state ceremonies to legally spread and perpetuate to other areas. Of course, the rulers can also benefit by better controlling the folk beliefs and maintaining the ruling order (Lian and Bian 2022, pp. 24–30).

Similarly, the cult of Lord Yan must be supported and rewarded by the ruler. This is not only the basis of its legitimacy to spread to other areas, but also an important condition for Lord Yan becoming a national god of water. Therefore, in the early Ming Dynasty, it is said that Lord Yan was crowned "The Waves-Calming Marquis" 平浪侯 for saving Ming Taizu in the water battle in Poyang Lake 鄱陽湖 (Pan 1975, p. 6). However, when we look at the history books of the Ming Dynasty, *Mingshi* 明史 (*History of Ming Dynasty*) (Zhang 1974), *Mingtaizu shilu* 明太祖實錄 (*Records of the Emperor Taizu)* (Dong et al. 1984), and *Daming huidian* 大明會典 (*Code of Great Ming Dynasty)* (Shen 1989), there is no written record of "Lord Yan being conferred a title for saving Ming Taizu", and it is not known when and why he was conferred the title.[5] The rumor that Lord Yan was named the "The Waves-Calming Marquis" by the Ming Dynasty (Lin 1989, p. 3539) eventually became the consensus of nationally unified history (X. Li 2017, p. 2387) and the writings of local prefectures and counties, behind which there must have been inevitable multidimensional efforts of social groups to promote Lord Yan's advancement to a national deity.

It is recorded in the Ming Dynasty's *Hongwu Jingcheng Tu Zhi* 洪武京城圖志 (*Collection of Geographical Information on Nanjing During the Hongwu Period*) that Lord Yan Temple is outside Dinghuai Gate 定淮門 (Wang 2018, p. 50), which offers a glimpse of the hidden logical connection between the cult of Lord Yan and the dynastic power system. In the early years of Hongwu, Dinghuai Gate was built and was originally named Maan Gate 馬鞍門. In 1374, the Gate was frequently flooded because of its location at a three-way intersection of the river and its facing the Qinhuai River 秦淮河. In the hope of controlling the river surface, the Gate was renamed Dinghuai (Chen 1999, p. 487). In the early Ming Dynasty, Lord Yan temple already existed in Nanjing City 南京城, although there was not yet any public construction of the temple and cult of Lord Yan at the state level, which invisibly influenced the folk society's cult of Lord Yan and the construction of temples (Snyder-Reinke 2009, pp. 2–50).

Lord Yan Temple in Qingjiang Town, Jiangxi Province, is the earliest ancestral temple in existence, so it is certain that the worship of Lord Yan was centered in the Qingjiang area and spread outward, as it was recorded that "the prestige of Lord Yan originated in the countryside, and became notable in Jiangxi, reaching far south to Hunan and far east to Jiangsu, so that mountains, valleys, rivers and sea all looked up to his lofty reputation, and expressed endless admirations" (Hu 1982, p. 154). Reading through the local histories of Hubei Province, we can see that most of the information on the cult of Lord Yan is scattered among the *Jianzhi zhi* 建置誌, *Cimiao zhi* 祠廟誌, and *Zazhi lei* 雜誌類. Because of the different compilation styles and content of local records, it is impossible to identify the main deities worshipped in some temples in the *Cimiao zhi*, the most common being "Shui Fu Temple" 水府廟 and "Shui Fu Shrine" 水府祠. As a result, the study focused on "Lord Yan Temple" and "Lord Yan Shrine" instead.

Along with Lord Yan's upgraded deity status and the increasingly significant manifestation of his power in the Ming–Qing era, the scope of the cult of Lord Yan in Jiangxi Province expanded outward. The neighboring Hubei Province, the rivers and lakes of which were densely intertwined, naturally became the first region where the cult of Lord Yan spread to the outside world.

In 1375, the monk Zhiyuan 智圓 (fl. 1370–1380) built Guanyin Pavilion 觀音閣 at the top of Chibi ji 赤壁磯, northwest to Hanchuan Gate 漢川門 in Huangzhou City 黃州城, which is also used to worship Lord Yan. In 1391, the Guanyin pavilion was incorporated into the Anguo Temple 安國寺 due to the local government's endeavor to clean up Buddhism (Lu 2017, p. 37). This shows that Lord Yan's worship had already spread from Jiangxi to Huangzhou Prefecture in the early days of Hongwu, the first region in Hubei Province to build a Lord Yan Temple.

In 1383, Xu Zhixian 徐志先 (late Yuan and early Ming) and other elderly people in Xiang yang County 襄陽縣 rebuilt the temple of Lord Yan five miles from Fancheng 樊城 in the north of the county (Zhang 2006, p. 188). According to this, Lord Yan Temple in Xiang yang County was built no later than the middle years of Hongwu. Similar to Huangzhou Prefecture, Xiang yang was also the region of Lord Yan Temple's early expansions in Hubei Province.

Then, the temples of Lord Yan were established one after another in Hubei Province. In 1386, Hu Xun 胡恂 (fl. 1380–1390), a citizen of Qizhou 蘄州, founded the temple of Lord Yan at the ruined site of Qianming Temple (Z. Wang 2017, p. 172). In the middle of Yongle 永樂 (r. 1403–1424), King Zhuzhen 朱楨 (1364–1424) of Chu built the temple of Lord Yan outside the south gate of Wuchang Prefecture (Y. Chen 2017, p. 436). During the Zhengtong 正統 (r. 1436–1449) period, Li Wei 李蔚 (fl. 1430–1450), the assistant of the County Magistrate in Qishui County 蘄水縣, built three Lord Yan temples in the east of Maqiao 麻橋, Lanxi 蘭溪 Town, and Bahe 巴河 Town in turn (Zhou 2017b, p. 158).

It is difficult to visualize the geographical and spatial layout of the Lord Yan temples in Hubei Province; thus, the spatial distribution of Lord Yan temples in Hubei Province during the Ming–Qing era was created based on the map of Hubei Province in the 25th year of the Qing dynasty (Figure 1).

According to the map, there were 20 temples of Lord Yan in the territory of Hubei Province in the Ming–Qing era, but the regional layout varied significantly.

The temples were mainly located in the central–eastern plain of Hubei Province, as well as in Huangzhou Prefecture and its counties.[6] This is because the locations are not far from Jiangxi Province, and merchants moved to Eastern Hubei in the Ming–Qing era.

Jingzhou Prefecture 荊州府, Wuchang Prefecture 武昌府, Xiangyang Prefecture, Hanyang Prefecture 漢陽府, and Jingmenzhou 荊門州 in the central and western parts of Hubei Province are full of rivers and lakes; therefore, natural disasters are frequent, and the worship of water gods is active. In this case, however, the distribution of Lord Yan temples is relatively small, which seems to be due to competition within the region between many traditional water gods with similar functions and Lord Yan.[7]

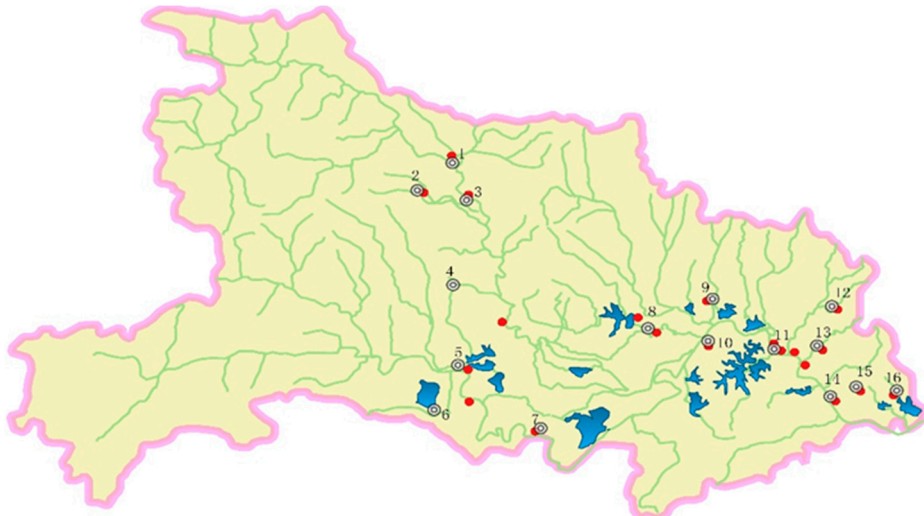

**Figure 1.** Spatial distribution of Lord Yan temples in Hubei Province during the Ming–Qing era. The numbers here represent the county administrative units where the Lord Yan temples are located, and the red dots represent the locations of Lord Yan temples: 1 = Xiangyang 襄陽, 2 = Nanzhang 南漳, 3 = Yicheng 宜城, 4 = Jingmenzhou 荊門州, 5 = Jiangling 江陵, 6 = Gong'an 公安, 7 = Jianli 監利, 8 = Hanchuan 漢川, 9 = Huangpi 黃陂, 10 = Jiangxia 江夏, 11 = Huanggang 黃岡, 12 = Luotian 羅田, 13 = Qishui 蘄水, 14 = Qizhou 蘄州, 15 = Guangji 廣濟, and 16 = Huangmei 黃梅.

In the western section of Hubei Province, there are no Lord Yan temples built in Shinan Prefecture 施南府 and Yichang Prefecture 宜昌府, but not because the regional community does not worship water gods. All prefectures and counties have built water temples, but people mainly worship the "Dragon King" 龍王 or "Dragon God" 龍神 (X. Wang 2017, pp. 501–2), with almost no foreign water gods.

Yunyang Prefecture 鄖陽府 worships Xiao Gong,[8] who is one of the local water gods in Jiangxi Province, just like Lord Yan. Although the exact reason is unknown, it is related to the choice of gods made by the local gentries or Jiangxi merchants.

In addition, the counties where Lord Yan temples are located in Hubei are distributed in the Yangtze River tributaries and lake regions, along where the water traffic is developed, highlighting Lord Yan's power of "helping with the water transport and protecting people in danger" as a water god.

Lord Yan serves as a water god in the region of rivers and lakes, and his worship, depending on the range of his powers and alternation of beliefs among other gods, could either become prosperous or decline. As a result, Lord Yan's temples are open to transformation or collapse.

During the Zhengde 正德 (r. 1506–1521) period, the Lord Yan temple in Qizhou County was destroyed by fire, but in 1526, it was rebuilt at the same place by the local gentries and later by the transport ambassador Chen Dazhong 陳大中 (fl. 1540) as the temple of Yu Wang 禹王 (Z. Wang 2017, p. 172). Both Yu Wang and Lord Yan were worshiped by the folk society as water gods, but the original intention of the transport envoy Chen Dazhong to replace Lord Yan with Yu Wang as the main god of the temple is not clear; the reason may be that either Lord Yan's power manifestation was inferior, so the local community chose to replace the deity,[9] or there was an official attempt to "standardize" (Watson 1985, pp. 292–324) the deity to suppress private worship.[10]

Lord Yan Temple in Yicheng County 宜城縣 was built on Guanzi Shore 灌子灘 of the Han River 漢水 twenty miles north of the county and later collapsed (Cheng 1975, p. 29). The collapse of the temple was apparently caused by years of disrepair, but the real reason may have been a decline in Han River water transport and merchant travel, or it may have been a comparable situation to Qizhou's case. During the Zhengtong period, the Lord Yan Temple in Maqiao, Qingquan Town 清泉鎮, whose construction was presided over by

Li Wei, the assistant of the County Magistrate in Qishui County, was combined with the Guan Wang Temple 關王廟 (Zhou 2017b, p. 158). The gods have very different functions, but share the same temple; the motive for this seems inseparable from the social tradition of multiple gods and the power–game dynamics among different hidden power groups.

However, the specific circumstances of the construction of the Lord Yan temples in Hubei Province are not known, so the local histories of Hubei Province are used as a basis to summarize the situation of the Lord Yan temples in Hubei Province during the Ming–Qing era (Table 1).

**Table 1.** Specific situation of the Temple of Lord Yan in Hubei Province during the Ming–Qing era.

| County | Quantity | Time | Location | Builders |
|---|---|---|---|---|
| Jiangling 江陵 | 1 | Early years of Hongwu | Shashi 沙市 | Unknown |
| Gongan 公安 | 1 | 9th year of Zhengtong | Ferry | Yu Yong 俞雍 (fl. 1430–1450, magistrate of a county) |
| Jianli 監利 | 1 | Unknown | Southwest of the county | Unknown |
| Jingmen | 1 | Unknown | Shayang 沙洋 | Unknown |
| Jiangxia 江夏 | 1 | Middle of Yongle | Outside the South Gate | King Zhuzhen of Chu |
| Hanchuan 漢川 | 2 | Unknown | East of the county Liu jia ge 劉家塥 | Unknown |
| Xiangyang | 1 | Unknown | Five miles from Fancheng in the north of the county | Xu Zhixian; Li Renyi 李人儀 (fl. 1450–1460, magistrate of a county) reconstruction |
| Nanzhang 南漳 | 1 | Unknown | Puji Bridge 普濟橋 on the east side of the county | Unknown |
| Yicheng | 1 | Unknown | North of the county 20 miles from Guanzi Shore | Merchant |
| Huangpi 黃陂 | 1 | Hongwu Period | One mile west of the county | Chen Zongying 陳宗英 (fl. 1368–1400, magistrate of a county); Sun Guan 孫冠 (fl. 1450–1465, magistrate of a county) reconstruction |
| Huanggang 黃岡 | 2 | 8th year of Hongwu | Chibi ji | Monk Zhiyuan |
| | | Unknown | Ruan family pavilion 阮家亭 in the city | Gentries and civilian |
| Huangmei 黃梅 | 1 | 48th year of Wanli 萬曆 (r. 1573–1620) | Duanjia zhou 段家洲 | Wang Keshou 汪可受 (fl. 1600–1630); Wang Fangchang 汪方長 (fl. 1630–1644) reconstruction |
| Luotian 羅田 | 1 | Unknown | One mile east of the county | Unknown |
| Qishui | 3 | Zhengtong period | Maqiao Lanxi Town Bahe Town | Li Wei (assistant to County Magistrate) |
| Qizhou | 1 | 19th year of Hongwu | Former site of Qianming Temple 乾明寺 | Hu Xun, citizen of Qizhou County); Gentries and civilian reconstruction |
| Guangji 廣濟 | 1 | Unknown | Unknown | Unknown |

According to the table, the time of construction of Lord Yan temples in Hubei Province was generally in the early years of the Ming

Dynasty and mostly concentrated in the Hongwu period. This reveals the prevalence of Lord Yan worship in Hubei Province in the early Ming period, coinciding with the time when the local communities in Jiangxi Province joined forces to attach themselves to the ruling class to enthrone Lord Yan.

Lord Yan temples are found in most of Hubei Province, especially in the eastern Huangzhou Prefecture, highlighting that Huangzhou Prefecture is the central area of Lord Yan worship in Hubei Province. To let Lord Yan bless the local community, vassal kings, grassroots officials, local gentries, civilians, and monks were all involved in the construction and repair of temples in the Hubei Province,[11] demonstrating the interactive evolution of multiple group forces and a social public space.[12]

In addition, some ancient temples of other types were transformed into Lord Yan temples. Similarly, some ancient temples of Lord Yan were also transformed into other types of temples. Although it is impossible to see whether there is a connection between other types of temples and Lord Yan temples or whether there is a connection between other gods and Lord Yan, it is possible to observe the evolution of religious beliefs in the local society through this behavior. For example, the Lord Yan temple in Qizhou County was destroyed by fire, and later it was rebuilt by local people and transformed by the transport ambassador Chen Dazhong into the temple of Yu Wang (Z. Wang 2017, p. 172). This behavior can be used even further to observe dynastic religious policies and social religion habits at different times.

The regional construction and space–time layout of Lord Yan Temples in Hubei Province cannot be separated from the religious habits and different groups' choices in the local society. Actually, this choice encompasses the religious habits of different social groups. Along with social changes, some temples of Lord Yan were merged, rebuilt, or destroyed in Hubei Province. This series of actions was a manifestation of different groups in local society competing for control of the gods (Faure 2007, p. 236). Of course, the process of different social groups competing for the gods is actually a manifestation of the confrontation between different levels of power in the local society (Liu 2011, pp. 7–11). For example, the Lord Yan temple in Huanggang County was built by local officials, gentries, and civilians, but after a dozen years, there was a struggle between different groups for ownership of the Lord Yan temple (Yu 2017, p. 160). Moreover, their purpose was to try to control local folk religion ideology and further dominate the local social order through the gods (Huang 2017, pp. 12–22).

## 4. Belief Identity and the Cross-Region Mobility of Lord Yan

The formation of folk god beliefs and believers has a certain regional character, whose circle of beliefs overlaps with the region formed with their geographical conditions, administrative divisions, and economic exchanges (Bol 2004). Since the Song–Yuan era, Lord Yan has been worshipped by Jiangxi's regional society. Through joint promotion by the government and the people in the early Ming Dynasty, the beliefs space was increasingly expanded to the outside, and Lord Yan became a water god with national influence.

Jiangxi and Hubei provinces, both belonging to the middle reaches of the Yangtze River society and sharing similar geographical and cultural characteristics, have relatively frequent regional exchanges, so Hubei Province has become the primary area for the spread of the cult of Lord Yan to the outside world. Since the change of the Yuan–Ming era, "the migration movement from Jiangxi to Huguang" 江西填湖廣 has been in full swing,[13] and the expansion of the foreign business and economy of "the merchant gangs of Jiangyou" 江右商幫 has been remarkable,[14] both of which contributed to the frequent geographical movement of people to a large extent.

During the Ming–Qing era, Lord Yan was accepted and worshipped for catering to the interests of the officials, gentries, and scholars of Hubei Province, gradually completed

the process of localization of folk beliefs, and was finally integrated into the historical construction of the local deity system (Zhang 2021, pp. 55–65).

During the transition between the Yuan and Ming eras, the society in the middle reaches of the Yangtze River experienced wars, resulting in a sparseness of people and desolation of the fields. When the Ming Dynasty was first established, the government's decree of immigration and Jiangxi's "land being full of people" 地滿人多 served as a dual impetus for the people of Jiangxi to migrate to Hubei on a large scale. During the Hongwu period, in Hubei, out of the 980,000 immigrants from various regions, about 690,000 were Jiangxi immigrants, accounting for 70% of the total population (Ge and Cao 1997, p. 148). Huangzhou Prefecture went through wars at the end of the Yuan dynasty, resulting in a huge reduction in the local population. To improve this situation, an immigration order that residents move to Huangzhou Prefecture was released at the beginning of the year Hongwu in the Ming Dynasty (Ge and Cao 1997, p. 130). Therefore, most of these Jiangxi immigrant families resided in the Huangzhou Prefecture. Starting in the middle of the Ming Dynasty, native people in Jiangxi flowed to Hubeu Province through the Ganjiang River and Poyang Lake waterway to avoid heavy taxation, which lasted until the Tianqi 天啟 (r. 1621–1627) period (Zhang 1995, pp. 16–17).

As more and more Jiangxi immigrants entered Hubei, the circle of the cult of Lord Yan expanded northward to the eastern part of Hubei Province due to the very similar physical geography of the two regions and the relatively close proximity between them.[15] Lord Yan Temple in Ruanjia Pavilion, Huanggang County, was jointly built by residents of a community, with the Ruan family playing a leading role (Dai 2001, p. 104). According to a survey, the residents of this community were mostly Jiangxi immigrants who responded to the government's orders in the early Ming Dynasty (Ge and Cao 1997, p. 129), and they were later codified in the household register of Huanggang County together to form Lijia 裏甲 (Mao 2017, pp. 121–29). Lijia was a grassroots unit of the Ming government ruling a local area.

In fact, Ruanjia Pavilion was a temple used to worship the ancestors of the Ruan family and later was also used by the residents of this area to offer sacrifices to Lord Yan (Anonymous 1996). Therefore, to a certain extent, Ruanjia Pavilion had become the center of worship for Lord Yan in this community. In addition, because the residents of this community all migrated from Jiangxi Province, they hoped to unite and improve their ability to resist various unknown risks by worshipping Lord Yan together. As immigrants from Jiangxi gradually integrated into local life, Lord Yan was also known to a growing number of social groups, further promoting the spread of the cult of Lord Yan to other areas in Hubei Province. The distribution of Jiangxi immigrants in Hubei Province decreased from east to west, about 80% of which was in the east of Hubei and on the Jianghan Plain 江漢平原, 60% in the north, and 30% in the northwest (Zhang and Mei 1991, pp. 77–109). The geographical layout of Lord Yan temples in Hubei Province was basically the same as the diffusion of Jiangxi immigrants, and they were relatively concentrated in the east of Hubei, sporadically distributed in the middle and the west. Jiangxi immigrants mainly entered Hubei Province northward along the Ganjiang River and Poyang Lake waterway and dispersed to various prefectures and counties westward and northwestward along the Yangtze and Han rivers through east Hubei (Shi and Zhang 1994, pp. 70–81). Lord Yan temples were mainly distributed along the main streams and tributaries of the Yangtze River, and the flow direction and location selection characteristics of the two obviously fit together, highlighting the intrinsic connection between Jiangxi immigrants and the flow of the cult of Lord Yan.

Due to the disturbance of the bandits at the end of the Yuan Dynasty, the Hubei indigenous people moved to Sichuan 四川 by the beginning of the Ming Dynasty. In the Ming–Qing era, Jiangxi immigrant groups became naturalized in Hubei Province, but still retained the cultural and customary traits of their original residence, so it is difficult to generalize the customs (Liu 2001, p. 29). In other words, the space of the cult of Lord Yan migrated with the people.

In the Ming–Qing era, the power of Jiangxi merchants was so huge that they could be found nationwide; as was recorded, "the land in Jiangxi is narrow and barren, even with hard work people cannot support themselves, so most of them have to go away to survive. Jiangxi is a traffic pivot where carriages and boats all come together, and people here mostly become merchants" (De 1970, p. 43). Wang Shixing 王士性 (1547–1598) in the Ming Dynasty revealed the reason for business: "The population is huge while the land is narrow in Jiangxi, Zhejiang and Fujian provinces, so people here are not able to support their families if they choose to stay at home instead of stepping out and giving full play of their skills. This situation is even more typical in Jiangxi" (Wang 1981, p. 80). People in Jiangxi had to become merchants to make a living, relying on the material basis of the increasing prosperity of handicrafts in commercial towns and the significant increase in the degree of commercialization of agriculture, thus promoting the expansion of the economic activities of Jiangxi merchants in a wider area.[16]

There are many merchant-led local Huiguan 會館 along the Yangtze River to the west and along the Han River to the northwest in Hubei Province (B. He 2017, pp. 60–71). The Jiangxi merchants living in Hubei Province, due to the need for friendship and industry standardization, established Jiangxi Huiguan in the prefectures and counties. At the same time, they set up local deities as spiritual models in the Huiguan for contacting fellow villagers and gathering local consciousness.

The Qingjiang area has the custom of business, so some merchants left their wives behind and walked thousands of miles to conduct business, while others settled down in other provinces, such as Guangdong 廣東, Jiangsu 江蘇, Yunnan 雲南, and Guizhou 貴州, but especially in Hubei, where most of them chose to stay (Qin 1642, p. 34). Due to these merchants originally coming from Qingjiang, residing in Hubei, and regarding Lord Yan as the hometown protection deity, their business routes often went through rivers and lakes. Therefore, some of the Jiangxi Huiguan would worship and offer sacrifices to the water God Lord Yan in the hope that he would help promote goodness in their hometown and bless the business road with safety (Oakes and Sutton 2010, pp. 260–93).

There were Jiangxi Huiguan in Jiangling County and Jianli County of Jingzhou Prefecture, in which the countryside and industry gods were worshipped (Ni 2001, pp. 58–61). Later, Jiangxi merchants were involved in presiding over the construction of Lord Yan temples in both counties. Lord Yan became the mutual spiritual support for the mutual assistance of Jiangxi merchants living in other provinces (Szonyi 2017, pp. 233–73). At the same time, local merchants also participated in the construction of Lord Yan Temples in the two counties (Ni 2001, pp. 58, 61). For example, Zhang Cheng 張誠 (fl. 1854) provided some funds during the construction of the Lord Yan Temple in Jiangling County, and Zhu Wen 朱文 (fl. 1854) provided land for the construction of Lord Yang Temple in Jianli County (Ni 1975, pp. 674, 702). To a certain extent, local merchants in Hubei were also an important group that worshiped Lord Yan.

In other words, Jiangxi merchants lived in Hubei Province to conduct business, vigorously built Jiangxi Huiguan dedicated to the local god Lord Yan, and participated in the construction of Lord Yan temples around Hubei Province, which undoubtedly made the cult of Lord Yan expand within the business region.[17]

It is stated in *Jiangxi tongzhi* 江西通志 (*General History of Jiangxi*) that "the worship of gods is common in all dynasties, and the temple's appearance should frequently be repaired and renovated, while the rites and ceremonies should follow the tradition" (Xie 1989, p. 1). Ceremonies and renovation activities, in which officials, gentries, and civilians all took part, were often held in the places where officials took office for the edification of civilians and to maintain grassroots society stability.

During the Ming–Qing era, the Lord Yan cult in Jiangxi spread to Hubei and was absorbed into the local deity system by the bureaucratic gentries and the civilian groups to meet their multiple interests. During the Hongwu period, Chen Zongying, a magistrate of Huangpi County, advocated the repair of the Lord Yan temple one mile west of the city together with other people (H. Li 2017, p. 157). Then, during the Tianshun 天順 (r. 1457–

1464) period, Sun Guan, the successor magistrate of Huangpi County, rebuilt it with others (H. Li 2017, pp. 157–58).

Chen Zongying and Sun Guan always emphasized the orthodoxy of Confucianism and were more repulsed by folk beliefs, so while they were governors of Huangpi County, they demolished the temples of many unknown deities (Yang 2017, p. 352). Along with the increasing influence of the cult of Lord Yan on Huangpi society, different social groups were actively involved in the construction activities of Lord Yan Temples (Yang 2017, p. 127). Therefore, the magistrates of the two counties had to work with other local groups to build Lord Yan Temples. In this way, they attempted to integrate the cult of Lord Yan into the local cultural system and further strengthened their control over the local community. In addition, the magistrates of the two counties were from other regions, and their workplaces were highly mobile (Yang 2017, p. 351); this feature objectively provided the possibility of spreading the cult of Lord Yan in a wider range.

A number of different situations emerged in Gongan County. In 1444, Yu Yong, magistrate of Gongan County, built the Lord Yan temple at the Ferry with other people (Wei 2017, p. 105). During the Longqing 隆慶 (r. 1567–1572) period, Qian Kuangzhi 錢匡之 (fl. 1568), magistrate of Gongan County, moved the Lord Yan temple to Zhongxuegang 中穴港 and transformed it into the Dongyue Temple 東嶽廟 with other people (Zhou 1975, p. 187). Dongyue temple was dedicated to the East Mountain Emperor 東嶽大帝, which is an orthodox deity in traditional Chinese society. During a flood disaster in the second year of Longqing, the East Mountain Emperor protected the gentries and civilians of Gongan County; however, Lord Yan did not play the role of protecting the local community (Zhou 1975, p. 445).

Thus, they decided to build a temple to worship the East Mountain Emperor. At the same time, with Lord Yan's influence diminishing in this region, his temple was transformed into Dongyue Temple. This indicates that different deities with the same function have competing characteristics in a region. The worship of deities by different social groups depends on whether the deity can function or not (Hansen 1990, p. 29), but there is no more evidence to support this claim.

The government, gentries, and civilians of Hubei Province being actively involved in the construction of the Lord Yan temples shows that the localization of the cult of Lord Yan completed the historical evolution of the folk deity worship system, and Lord Yan became a local genealogical deity worshiped by multiple communities. The specific construction date of the Lord Yan temple five miles from Fancheng, north of Xiangyang County, is unknown. In 1383, the temple was rebuilt by Xu Zhixian and others. In 1459, it was rebuilt by Li Renyi and others (Zhang 2006, p. 188). Later, both Lord Yan and Xiao Gong were worshiped in this temple, and it became well-known for the gods responding to people's wishes (E. Chen 2017, p. 535).

In 1620, Wang Keshou built the Lord Yan temple at Duanjia zhou in Huangmei County, Huangzhou Prefecture 黃州府, which was later renovated by Wang Fangchang (Jia 2017, p. 50). According to Wang's genealogy, the Wang family's ancestors moved from Huizhou Prefecture 徽州府 to Huangzhou Prefecture to escape the war in the late Song Dynasty, and members of the Wang family also earned their living mainly from agriculture (Anonymous 1945). Therefore, the Wang family worshipped Lord Yan, probably hoping that Lord Yan would bless their agricultural activities.

In the Ming Dynasty, many members of the Wang family took part in the imperial examinations 科舉考試 and obtained several titles; for example, Wang Keshou acquired the status of a scholar 秀才, and Wang Fangchang acquired the status of a Gongsheng 貢生 (Anonymous 1945). In the meantime, they offered sacrifices to Lord Yan and thanked Lord Yan for helping them win the title. This shows that the connotation of Lord Yan's faith expanded. Although it is not known why the Wang family worshiped Lord Yan, the Wang family repaired the temple from generation to generation, revealing the close relationship between Lord Yan and the Wang family.

As a water god who blesses the safety of ships running between rivers, lakes, and seas, the spatial scope of Lord Yan's cult spread outward and was closely linked to water practitioners. The Lord Yan temple, built on the Guanzi shore of Han River, 20 miles north of Yicheng County, was built a long time ago, but it is not known when it was built or who built it (Hao 2001, p. 273). The Guanzi shore was a busy ferry on the Han River, where a number of merchants, sailors, and fishermen converged. By offering sacrifices to Lord Yan, they hoped to ensure the safety of water transportation and further obtain material wealth; therefore, the Lord Yan Temple was continuously repaired (Cheng 1975, p. 275).

Of course, the fishermen were mainly from Yicheng County, which is in line with the annals of Yicheng County, wherein a considerable number of local people engaged in fishing activities (Yao 1975, p. 19). However, the specific identities of the merchants and sailors cannot be known from the available information. It is certain that they played an important role in spreading Lord Yan's faith. In addition, the group of water practitioners was diverse and complex, but they all believe that Lord Yan protected the economic production of fisheries and the safety of water transportation.

The Shi family ancestors in Huanggang County moved from Jiangxi to Hubei to escape the war in the late Yuan Dynasty (Anonymous 1988). Shi Gulu 石穀祿, the ancestor of the Shi family, was an official, but resigned for some unknown reasons. After returning to his hometown, he started to engage in business activities. Once, when he went to the Jiangxi area for business, he was protected by Lord Yan when he was in danger onboard.[18] After he returned home, he built a statue of Lord Yan and worshipped him in his own house. At the same time, he vowed that he would tell his descendants to worship Lord Yan forever. To a certain extent, Lord Yan had become the protector of the Shi family. With the growing number of Shi family members, they decided to jointly fund the construction of a Lord Yan temple, and the main organizers of the project were Shi Changcai 石昌才 (fl. 1552), Shi Shengfu 石勝富 (fl. 1552), Shi Shengchu 石勝楚 (fl. 1552), Shi Chengmei 石成美 (fl. 1552), Shi Chongyou 石崇又 (fl. 1552), Shi Qinghe 石慶和 (fl. 1552), and Shi JinXing 石錦興 (fl. 1552) (Anonymous 1988). At the same time, according to the Shis' genealogy, a common feature of these family members is that they were all engaged in business activities (Anonymous 1988). This seems to indicate that the Shi family evolved into a commercial family, but there is no more evidence to prove it. As a result of their commercial activities, the cult of Lord Yan was further spread to other regions. More importantly, the whole Shi family formed a cultural community centered on the cult of Lord Yan, and the Lord Yan temple also became the center of power for this family community (Freedman 1965, pp. 82–93).

The cult of Lord Yan grew among the socially diverse groups of Hubei Province; the temples were built to worship him at all major river and waterway traffic routes, which strongly promoted the spatial expansion of the cult of Lord Yan.

## 5. Conclusions

During the Yuan Dynasty, Lord Yan was worshipped by local communities as a regional water deity for a long time, and the beliefs space was limited to a corner of Jiangxi society. At the beginning of the Ming Dynasty, the cult of Lord Yan flowed outward due to the text construction of pluralistic groups (Han 2015, pp. 86–96, 220) and the instillation of official consciousness by the ruling class (transforming the content, incorporating the rituals, granting the name, and giving the temple title) (Zhu 2008, p. 178); Lord Yan, thus, was changed from a local water deity who sheltered the Qingjiang counties into a national water deity who was responsible for calming the wind and waves, as well as guaranteeing safe navigation.

The specific time of the cult of Lord Yan's spread from Jiangxi to Hubei cannot be confirmed, but the construction of Lord Yan temples was mostly concentrated in the Hongwu period of the Ming Dynasty, which seems to indicate that the cult of Lord Yan entered and flourished in Hubei no later than the Ming Taizu period. The spatial layout of ancestral temples in Hubei Province differs significantly due to the different degrees of regional be-

liefs, which are concentrated in the eastern Huangzhou Prefecture, while the Huangzhou Prefecture is the central area of beliefs in Hubei Province. The sparse distribution of temples in the central and western regions of Hubei Province seems to be the result of the selection between gods and similar social functions in the region. At the same time, the counties with temples are all located along the tributaries of the Yangtze River and lakes with well-developed waterway traffic, highlighting the functional characteristics of the water god.

During the Ming–Qing era, the flow of immigrants from Jiangxi to Huguang; the cross-regional commerce of merchants from Jiangxi; the acceptance and worship of officials, gentries, and scholars; and the acquiescence and use of the ruling class together constituted the multidimensional motivation for the outward expansion of the cult of Lord Yan.

In the process of spreading the cult of Lord Yan in Hubei Province, the Vassal kings, officials, gentries, merchants, and civilians attempted to gain private benefits by actively participating in the rituals, temple constructions, and repair activities of Lord Yan, and the water practitioner group was especially committed to Lord Yan. They hoped that the god Lord Yan would bless the place and protect the people in danger; the process of human–god interaction can be regarded as a "gift exchange",[19] along with Lord Yan's growing manifestation of powers. Thus, the scope of the belief's community increasingly expanded, and they all looked up to its heroic wind on the water, prompting the spread of the cult of Lord Yan to a wider area.

The process of spreading the cult of Lord Yan in Jiangxi to Hubei is different from the flow of gods in southern China society (He and Faure 2021, pp. 181–205). Meanwhile, the change in deity status (deity birth) and the flow of specific deities across borders (religious traditions) also differ (Lu 2013, pp. 33–52). However, they all belong to a cultural-shaping process within the regional society.[20]

The social groups construct the content of the cult of Lord Yan and try to integrate cultural consciousness in order to advance into the national ritual system, but it is not a "Ritual Signs" in the fully orthodox sense.[21] Rather, it belongs to a structural process, in which different gods' traditions and subjective perceptions of social groups intertwine. In short, this process concentrates on the expression of various aspects of different people's thoughts, the characteristics of regional social life, and the identity of groups within the state during the historical period (Zhao 2018, pp. 1–11, 193).

The regions formed by the propagation of deities are based on physical geography, administrative units, and economic space, but may also break through such limits to form a larger religious region (Pi 2008, p. 253). The flow and expansion of the Yan Gong faith are more like the transfer of religious consciousness from the Qingjiang area to other places. Through the fusion of different religious beliefs afterward, it is integrated into the structure of other regional belief systems. This process would not be possible without the cooperation of multiple social groups with similar motivations to create a standardized combination of rituals within the cultural structure (Watson 2003, pp. 98–114).

By observing the spread of the cult of Lord Yan from Jiangxi to Hubei and the state of the cult of Lord Yan in Hubei, we can consider this a method of regional cultural continuity. In other words, only when the folk gods are continuously worshiped by the cult of different social groups and spread to other areas can they continue to exist in society. In the process, the connotation of the cult of Lord Yan was also integrated into some local knowledge and cultural elements (Geertz 2014, pp. 38–44), so the cult of Lord Yan is commonly recognized by different social groups or social organizations in Jiangxi and Hubei provinces. For example, after Lord Yan was spread to Hubei as the god of water in Jiangxi, it was added to the function of blessing agricultural harvests. In a deeper sense, the cultural order and social relations of grassroots society can be observed through the state of the cult of Lord Yan by different groups and regions.

**Author Contributions:** Conceptualization, S.Z. and H.S.; methodology, S.Z.; software, H.S.; validation, S.Z.; formal analysis, H.S.; investigation, S.Z.; resources, S.Z.; data curation, H.S.; writing—

original draft preparation, S.Z.; writing—review and editing, S.Z.; visualization, S.Z.; supervision, H.S.; project administration, H.S.; funding acquisition, H.S. All authors have read and agreed to the published version of the manuscript.

**Funding:** This research received no external funding.

**Data Availability Statement:** The main data in the article are from local chronicles and field research.

**Conflicts of Interest:** The authors declare no conflict of interest.

## Notes

[1]  At present, some of the research results of Lord Yan in academic circles include (Rao 2009), (Song 2014, pp. 118–23), (Tang and Zhang 2013, pp. 253–59), (Hu 2015, pp. 11–14), (Cheng 2016, pp. 127–34), and (Wang 2020, pp. 105–14).

[2]  The original publication of the *Chu Yin Ji* 樗隱集 collection is no longer in circulation. Jiao Hong 焦竑 (1540–1620) *Guoshi jingji zhi* 國史經籍誌 of the Ming Dynasty vigorously included the anthologies of the Yuan–Ming era, but not this one. During the Qianlong 乾隆 (1736–1795) period, the government compiled the *Si ku Quanshu* 四庫全書, collected a number of poems and articles from the *Yongle Da Dian* 永樂大典, and re-edited them into the six volumes of *Chu Yin Ji*, which were compiled into the category of other collections.

[3]  The Ming rulers themselves were adept at creating gods. "Lord Yan, who was produced in Song–Yuan era and was a local water god in Jiangxi, was made the 'the Waves-Calming Marquis' for blessing Zhu Yuanzhang's victory in the battle of Poyang Lake". The cult of Lord Yan was reinforced by Zhu Yuanzhang and worshipped nationwide, so it seems certain that it arose before the Ming Dynasty (the middle and late Yuan Dynasty) (Zhao 2017, p. 13).

[4]  Lord Yan's real name is Beihai 北海, a native of Yanfang 晏坊 during the Yuanyou 元祐 (1086–1094) period of the Song Dynasty. Wherever there were locusts, floods, or droughts, the people prayed to the gods for blessings, and they were answered. The other is Yan Wuzai in Qingjiang Town, 30 miles north of Qingjiang County, Linjiang Prefecture (Tan 2006, p. 510). The two Lord Yans are worshipped by many counties today (Tan 2006, p. 510). In short, there is no uniformity in the origin story of Lord Yan, or Yan Wuzai of the Yuan Dynasty period, or Yan dunfu 晏敦複 of the Song Dynasty; additionally, there are said to be the Sunwu 孫吳 (a dynasty in China, 229–280) before people, and so on (Zong and Liu 1986, p. 356).

[5]  One theory is that Lord Yan was made"the Waves-Calming Marquis" in the early Hongwu period of the Ming Dynasty (Cui 2017, p. 326). Another theory is that Lord Yan was made "the Waves-Calming Marquis" in the middle of the Yongle period of the Ming Dynasty (Shi 1989, p. 459).

[6]  In the Ming and early Qing dynasties, Huangzhou Prefecture was under the administration of "Huguang buzhengshisi" 湖廣布政使司, and had jurisdiction over one state and eight counties. In the seventh year of Yongzheng 雍正 (1722–1735) in the Qing Dynasty, Huangpi County was transferred to the jurisdiction of Hanyang Prefecture. Therefore, Huangzhou Prefecture had jurisdiction in one state and seven counties and was transferred to the jurisdiction of "Hubei buzhengshisi" (Ying 1976, p. 3).

[7]  There are many water gods in the central and western regions of Hubei Province, mainly including natural water gods (for example, River Gods and Juzhang 沮漳 Gods), personified water gods (for example, Yang Hou 陽侯, Xiang Jun 湘君, Xiang Furen 湘夫人, Qu Yuan 屈原, Da Yu, Liu Yi 柳毅, Zhao Yu 趙昱, Xu Jingyang, Liu Qi 劉琦, Tian Fei, Xiao Gong, and Lord Yan), and mythical animal water gods (for example, Dragon Gods). Each water god is treated differently by local officials and people because of its own spirituality and different degrees of influence, and the development of the cult of the water god is geographically unbalanced (Yuan 2019, pp. 9–28).

[8]  Xiao Gong is a famous water god of Jiangxi in the Ming–Qing era, and his worship was formed in Xingan County 新淦縣, Jiangxi Province, during the Yuan–Mingera. The character archetype of the water god Xiao Gong mainly includes three generations of the Xiao family's grandchildren (Xiao Boxuan 蕭伯軒, Xiao Xiangshu 蕭祥叔, and Xiao Tianren 蕭天任), and the social group gradually took Xiao Tianren as Xiao Gong (Zhou 2017a, pp. 496–97, 500).

[9]  Chinese folk religious traditions are mutually exclusive, and the people sought protection from many deities from different religious traditions at the same time. Additionally, their choices are determined by divine power rather than by a particular religion they belong to (Hansen 1990, p. 29).

[10]  By imposing a will or language on local deities, the state tried to intervene in local society in a cultural way (Averill 2006, p. 58).

[11]  Local officials, gentries, and civilians were involved in the maintenance and management activities of the temple, which inevitably promoted the evolution of the function of beliefs space towards diversification and made the temple a site of a multidimensional power competition (Liu and Zhang 2020).

[12]  State power and social forces present a complex pattern of competition or collaboration in local societies. In particular, different forces in local public affairs have shown great autonomy and dynamism. Multiple group forces jointly shape public space, and public space inversely influences multiple group forces (Wu 2009, p. 5).

[13]  Since the Ming–Qing era, local county records and private genealogies have often recorded the saying "the migration movement from Jiangxi to Huguang", but this is not found in official political books and private national histories and is mostly referred to by folk society (Xia 2013, p. 220).

[14] Fang Zhiyuan's 方誌遠 comprehensive discussion of the rise of "the merchant gangs of Jiangyou", the scope of activities, modes of operation, capital composition, and social composition basically forms a general overview of "the merchant gangs of Jiangyou". However, there is no unified opinion among academics on the time of the rise of "the merchant gangs of Jiangyou" (Fang 1995; S. He 2017).

[15] Belief circles are voluntary organizations formed by regional believers, centered on the cult of a God (Lin 1990, pp. 41–104).

[16] With Jiangxi's commercial towns and rural commodity markets, the Merchant Gangs of Jiangyou became increasingly prosperous in the mid-Qing period. At the same time, with the increasingly large commercial organization of the Merchant Gangs of Jiangyou, its geographical scope has expanded to the whole country. In addition, to a certain extent, the Merchant Gangs of Jiangyou can be considered the product of Jiangxi's exiles during the Ming–Qing Era (X. Zhang 2015).

[17] The total number of commercial Huiguan around Hubei Province is 295, and the number of Jiangxi Huiguan is 66, accounting for about 22.4% (Zhang 1995, pp. 287–91).

[18] Reference: 石氏建修晏公廟碑記 (Records of the Construction of the Temple of Lord Yan by Shi), author: Shi, Yaochun 石耀春 (fl. 1889–1904). It now exists in Lord Yan Temple in Sandian Town 三店鎮 (originally part of Huanggang County), Xinzhou District 新洲區, Wuhan City 武漢市.

[19] Both gift giving and gift returning are obligatory. Diverse social groups worship Lord Yan and build temples in the hope that the gods will bless them. Lord Yan enjoys the worship of the social group, so he must exert his power to protect all the people (Mauss 2016, p. 5).

[20] The operating principle of the regional social order is not a rigid theoretical law, but the overall mobilization mechanism of how the regional society is organized, which is the historical practice of mobile and living people (Zhang 2011, pp. 171–88, 223–24).

[21] Local liturgical practices are based on the concept of orthodoxy, and when different orthodox traditions collide, they create an overlap of rituals (Faure and Zhang 2016, pp. 21–23).

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
