# Peer review of "Mobility to Other Locations: A Study on the Spread of the Cult of Lord Yan from Jiangxi to Hubei in the Ming–Qing Era"

_religions, doi:10.3390/rel14050593_

Round 1
Reviewer 1 Report
This is an interesting article, based on compelling archival evidence, which deserves to be published.
However, please make sure that the academic English is checked carefully by a native speaker of academic background. Secondly, there are some recurring terms which should be changed, e.g. Yuan / Ming *era* instead of "dynasty".
Some more Western authors should be referred to, especially Vincent Gossaert's "Becoming a God".
Please re-organise your bibliography in alphabetical order.
Otherwise a very good article!
Author Response
Thank you very much for your professional review work on our article. Please see the attachment.

Reviewer 2 Report
This article is a careful study of the spread of the cult of Lord Yan 晏公 from Jiangxi to Hubei (esp. Huangzhou Prefecture in the eastern part of the province) in the early decades of the Ming and the driving forces behind the spread. While a few articles were published on the topic in Chinese, the cult has thus far received little attention in the English-language scholarship. The article is well organized. The biggest problem with the article is the language: Grammatical mistakes and unidiomatic expressions are plenty, which presents a significant barrier for understanding. The language issue also seems to be the cause of improper use of important analytical concepts such as the “civil society.” Sometimes the Chinese phrases (e.g., 甕 [urn? jar?], 平浪侯 [the Waves-Calming Marquis?], 臨江路 [Lingjiang ROUTE] as a Yuan-dynasty administrative unit) are untranslated, not fully translated, or incorrectly translated. At other times, it is difficult to understand what the authors mean (e.g., “geographical society,” “runs through the territoriality of the inherent social order”). Apparently because of these language barriers, I find the analytical passages (the top two paragraphs on p.9 and the last paragraph on p.13) nearly incomprehensible. The author must work intensively with an English-language editor, hopefully someone familiar with Chinese history, to solve these language issues. I suggest that Yangong 晏公 be translated as “Lord Yan.”
The author develops his/her argument mainly in Sections 3 and 4. Section 3 explores where the Lord Yan temples were located in Hubei. Section 4 explores the causes behind the spread of the cult from Jiangxi to Hubei in the early Ming.
Section 2 actually handles a different set of topics: the origin of the cult, the evolving and divergent interpretations of the deity. This section provides a useful context (the origin and evolution of the cult) for discussion in the subsequent sections. Some of the ideas in this section (enfeoffment of the deity—or at least a belief that he was enfeoffed and recognized by the state—and stories that tied the god to the Ming legitimacy) are relevant to later discussions on the spread of the cult: the author seems to suggest that state acceptance of the cult was a factor behind the spread of its influence (“after the joint promotion by the government and the people in the early Ming dynasty, the belief space was increasingly expanded to the outside” [p.9]). Nonetheless, I find this section very different to follow. Language, once again, was what got in the way, but more importantly, I think this section is largely narrative and not argumentative. If the author intends to make one or more arguments here, these arguments are buried in the telling of miraculous stories.
So, I think it would be best if the author significantly cuts down Section 2. Instead of presenting all the evidence and narrating all the stories, the author should aim at provide a concise summary of the origin and evolution of the cult in Song, Yuan, and early Ming that set the stage for the actual focus of this paper: the spread of the cult to Hubei and what caused the spread. The evidence presented in Section 2 indicates that there are other interesting lines of inquiry to pursue, but they have to be dealt with in a different paper. (Some of the stuff in Section 2 can be included in later discussions. I will get to this below.)
By cutting down on Section 2, the author will be able to expand Section 4. Section 4 explores the driving forces behind the spread of the cult and what accounts for the concentration of its temples in Huangzhou, but it leaves many questions unanswered. In my view, there were two different types of factors behind the spread. First, similar economic environments and state acceptance are necessary conditions that made the spread possible. Second, there were historical agents who actively “carried” the cult from Jiangxi to elsewhere and/or agents who actively “adopted” an outside cult. Who were these agents? Discussion in Section 4 brings up several social groups: the Jiangxi immigrants, the Jiangxi merchants and their guild halls (會館?), local families (e.g., the Wang 汪 and the Shi [p.11]), and perhaps local officials too. These discussions are where the strengths of this article lies, but the discussion is inadequate (Compared to Section 2, very few sources are cited and analyzed in detail) and leaves many questions unanswered. Here are a few that I think should be further explored:
1. It appears that the Jiangxi immigrants played a critical role in the spread of the cult. Did Jiangxi immigrants formed distinctive local communities of their own (something like the 眷村 in Taiwan) and separate from indigenous Hubei people in the early Ming? When the cult of Lord Yan spread to Hubei, was it practiced exclusively or mainly by the Jiangxi immigrants, or also by the indigenous Hubei folks? One may get a sense of this by looking carefully at the stele inscriptions (碑記) and see who advocated for and funded the construction of temples. The bigger issue at stake here is: To what extent was the cult an expression of a distinctive identity of the Jiangxi immigrants?
2. The author also suggests that occupation also played a role. The cult was particularly popular among those whose livelihood had to do with water. Merchants, for sure. But what were the other occupations of Lord Yan’s adherents in the historical sources? “Water practitioners” does not mean anything in English. It is important to be accurate and specific here: boatmen? Fishermen? Or else?
3. To what extent do these “water folks” overlap with Jiangxi immigrants? In other words, did native Hubei merchants also worship Lord Yan, etc.?
4. The role of local families: The author mentions the strong connections between local families, such as the Wang and the Shi, and the cult. Were these families Jiangxi immigrants too? We are told that the Shi were involved in commerce, but what about the Wang?
5. What was the role of local officials? We see that local officials played an active role in building and renovating these temples (paragraph 3 on p.11), but the author seems to suggest that these officials did not take initiative and were simply responding to local demands. It may well have been so, but the reader would love to see the evidence for this judgment.
6. Did the Daoist and Buddhist clergy play a role in the spread of the cult at all? (The discussion of Zhiyuan on p.5 seems to suggest so. Or was it an exceptional case?)
In pursuing these questions, I think the author will be able to engage more meaningfully and in greater depth with a large body of scholarship that discusses 1) the relationship between folk religion and identity (e.g., Sangren on Taiwan), and 2) the role of the state, merchants, and scholar-officials in the spread of a cult (e.g., James Watson on Mazu, Pi Qingsheng 皮慶生, Valerie Hansen, etc.).
On a side note: The author summarizes that “The belief regions formed by the propagation of deities are based on physical geography, administrative units and economic space, but may also break through such limits to form a larger religious region” (p.13). What the discussion in the paper actually shows that physical/economic geography and the movement of people mattered, but the borders of “administrative units” had no role in the spread of a cult (these borders were not barriers or facilitators).
“Notes” should be numbered. “References” should be better formatted (e.g., italics for book titles) and listed by the last names of authors.
Author Response

(The authors gave the same response as above.)
